# Two distinct domains contribute to the substrate acyl chain length selectivity of plant acyl-ACP thioesterase

Fuyuan Jing[1,2,3], Le Zhao[4], Marna D. Yandeau-Nelson[2,5] & Basil J. Nikolau[1,2,3]

The substrate specificity of acyl-ACP thioesterase (TE) plays an essential role in controlling the fatty acid profile produced by type II fatty acid synthases. Here we identify two groups of residues that synergistically determine different substrate specificities of two acyl-ACP TEs from *Cuphea viscosissima* (CvFatB1 and CvFatB2). One group (V194, V217, N223, R226, R227, and I268 in CvFatB2) is critical in determining the structure and depth of a hydrophobic cavity in the N-terminal hotdog domain that binds the substrate's acyl moiety. The other group (255-RKLSKI-260 and 285-RKLPKL-289 in CvFatB2) defines positively charged surface patches that may facilitate binding of the ACP moiety. Mutagenesis of residues within these two groups results in distinct synthetic acyl-ACP TEs that efficiently hydrolyze substrates with even shorter chains (C4- to C8-ACPs). These insights into structural determinants of acyl-ACP TE substrate specificity are useful in modifying this enzyme for tailored fatty acid production in engineered organisms.

[1] Department of Biochemistry, Biophysics, and Molecular Biology, Iowa State University, Ames, IA 50011, USA. [2] Center for Biorenewable Chemicals, Iowa State University, Ames, IA 50011, USA. [3] Center for Metabolic Biology, Iowa State University, Ames, IA 50011, USA. [4] Department of Chemical and Biological Engineering, Iowa State University, Ames, IA 50011, USA. [5] Department of Genetics, Development and Cell Biology, Iowa State University, Ames, IA 50011, USA. Correspondence and requests for materials should be addressed to B.J.N. (email: dimmas@iastate.edu)

Fatty acids (FAs) are some of the most energy-dense molecules that biological systems can produce. Therefore, there is increasing interest in harnessing the FA biosynthetic machinery for bioenergy and bioproduct applications[1,2]. The biosynthesis of FAs is catalyzed by FA synthases (FAS) that occur in two ternary forms, the multicomponent type II FAS that occurs in bacteria and plants[3], and the multifunctional type I FAS that occurs in fungi and animals[4]. The industrial application of FAs is determined by two attributes, their carbon chain length and the degree of unsaturation. The former attribute is enzymologically determined by the substrate specificity of the acyl-ACP thioesterase (TE) that catalyzes the terminal reaction of the type II FAS system. TE catalyzes the hydrolysis of the acyl-ACP intermediates of the iterative FAS pathway, to release the free FA and thus terminates the process of FA biosynthesis. Therefore, acyl-ACP TEs play a crucial role in determining the product profile of type II FAS, specifically the carbon chain length of the FA products.

The biotechnological focus on acyl-ACP TEs was primed by the discovery that this enzyme is the major determinant that enables seeds of certain plants (e.g., California Bay Laurel tree, and members of the Palmae family) to produce laurate-rich oils[5,6]. This trait was subsequently transgenically transferred to annual crops, such as oilseed rape, resulting in the accumulation of seed oil containing over 50% of lauric acid[7]. Furthermore, over-expression of these TE enzymes in *Escherichia coli*, provides these cells with an additional "product sink" for FAS, which results in the overproduction of FAs[8]. This latter, relatively simple metabolic engineering accomplishment, spurred the development of microbial biocatalytic systems for the biochemical reduction of oxidized-carbon feedstocks (e.g., $CO_2$, sugar, or lignocellulose) to fuel-like molecules, such as FAs[1,2].

Initial characterizations of plant-sourced acyl-ACP TEs led to their classification as either FatA or FatB TEs[9]. FatA TEs specifically hydrolyze unsaturated acyl substrates (i.e., 18:1-ACP), and FatB TEs preferentially act on saturated acyl-ACPs with acyl chains varying from 8 to 18 carbons in length. More recent reclassifications of a broader spectrum of acyl-ACP TEs sourced from both plants and bacteria revealed that the FatB-type TEs can be further categorized into three subclasses. Subclass I FatB TEs prefer 14- and 16-carbon acyl-ACPs, and these have been hypothesized to be the evolutionarily ancient and ubiquitous TE enzymes[9]. Subclass III FatB TEs act predominantly on 8-carbon acyl-ACPs; the subclass II FatB TEs display broader substrate specificities, spanning from 8 to 16 carbons[10].

In the context of this more recent knowledge, we took advantage of the greater opportunity to utilize the power of bioinformatic structural analyzes and diverse TE sequences with distinct substrate specificities to identify domains and specific amino-acid residues that determine the substrate specificity of the FatB class of acyl-ACP TEs. Specifically, we utilized two FatB-type acyl-ACP TEs from *Cuphea viscosissima* (CvFatB1 and CvFatB2) that share more than 70% sequence identity but express distinct substrate specificities: CvFatB1 belongs to subclass III FatB TEs and primarily acts on C8/10-ACPs; CvFatB2 belongs to subclass I FatB TEs and primarily acts on C14/16-ACPs[10]. Two complementary strategies, domain-swapping and site-directed mutagenesis, were used to identify 11 residues that affect the substrate specificity of the two acyl-ACP TEs. In addition, these characterizations led to the creation of more than 60 synthetic enzymes, some of which have acquired catalytic capabilities that may have relevance to the more efficient conversion of sugar-derived carbon to energy-dense molecules that have applications as biofuels or bioproducts.

## Results

### Substrate specificity of chimeric acyl-ACP TEs. Figure 1a, c illustrates that CvFatB1 and CvFatB2 share more than 70%

sequence identity, but when expressed in *E. coli*, CvFatB2 acts on 14- and 16-carbon fatty acyl-ACPs, whereas CvFatB1 acts on 8- and 10-carbon fatty acyl-ACPs[10]. An integrated two-step strategy was used to identify the subset of the ~80 residues that differ between these two proteins, which determine this difference in substrate specificity.

Initially, domain-swapping experiments were used to map the location of critical residues that distinguish the substrate specificity between CvFatB1 and CvFatB2. Six overlapping fragments of approximately equal length were used to generate chimeric acyl-ACP TEs (Fig. 1b). In the first round of this experiment, six chimeric acyl-ACP TEs were generated by domain-swapping of three fragments, I–II, III–IV, and V–VI (rTE3, rTE12, rTE15, rTE48, rTE51, and rTE60; Fig. 1c).

Each chimeric TE, and the parental CvFatB1 and CvFatB2 TEs were expressed in *E. coli* strain K27, and the in vivo FA productivity and substrate specificity of the resulting strains were compared (Fig. 1c). Swapping fragments V and VI, as in rTE3 and rTE60, had no effect on the substrate specificity as compared to the parental TEs (Fig. 1c). In contrast, each of the other first-round chimeric TEs that harbor substitutions of fragments I to IV (i.e., rTE12, rTE15, rTE48, and rTE51), exhibit altered substrate specificities as compared to the parental CvFatB1 and CvFatB2 enzymes (Fig. 1c). Therefore, the residues that determine the different substrate specificities of the two enzymes reside within these four N-terminal fragments, and not in fragments V and VI.

A second round of domain-swapping individually swapped fragments I–IV, generating an additional 12 chimeric TEs. The seven chimeric constructs that contain fragment III from CvFatB2 (i.e., rTE8, rTE24, rTE28, rTE40, rTE44, rTE56, and rTE60) exhibit a FA profile that is similar to the parental CvFatB2 (Fig. 1c). In contrast, the six chimeric constructs that contain fragment III from CvFatB1 (i.e., rTE4, rTE16, rTE20, rTE32, rTE36, and rTE52) generate FA profiles that are similar to the parental CvFatB1. The importance of fragment III in determining substrate specificity of the two TEs is clearly demonstrated by comparing the FA profiles of rTE8 with CvFatB1, and comparing rTE52 with rTE60. rTE8 differs from CvFatB1 only by the composition of fragment III, which is CvFatB2-derived, and yet exhibits a FA composition that is very similar to that generated by CvFatB2. Conversely, the single switch of fragment III from a CvFatB2 sequence (as in rTE60) to a CvFatB1-derived fragment III (as in rTE52) completely alters the substrate specificity of the chimera from a CvFatB2-like to a CvFatB1-like enzyme. These results therefore establish that fragment III-residing residues are most critical to the substrate specificity difference of the two TEs.

### Computational predictions of substrate-specifying residues. The second strategy for identifying the residues that distinguish the substrate specificity of CvFatB1 from CvFatB2 compared these two sequences to a larger set of 21 acyl-ACP TEs, whose substrate specificities had previously been characterized[6,10–12]. Fourteen of these acyl-ACP TEs belong to subclass I FatB TEs, and the other seven belong to either subclass II or III FatB TEs. Based upon these two classifications, we examined the nature of the distinguishing residues, and hypothesized whether each has a potential role in determining substrate specificity according to the following two criteria: (1) a residue that is conserved only in one subclass would indicate an important role in determining the substrate specificity of that subclass; and 2) a residue that differs between the two subclasses in the chemical nature of its side chain (i.e., size, charge, and hydrophobicity) would indicate that it may contribute to the difference in substrate specificity between the subclasses.

Among the 80 residues that distinguish CvFatB1 from CvFatB2, 4 residues absolutely fit these criteria (i.e., residues

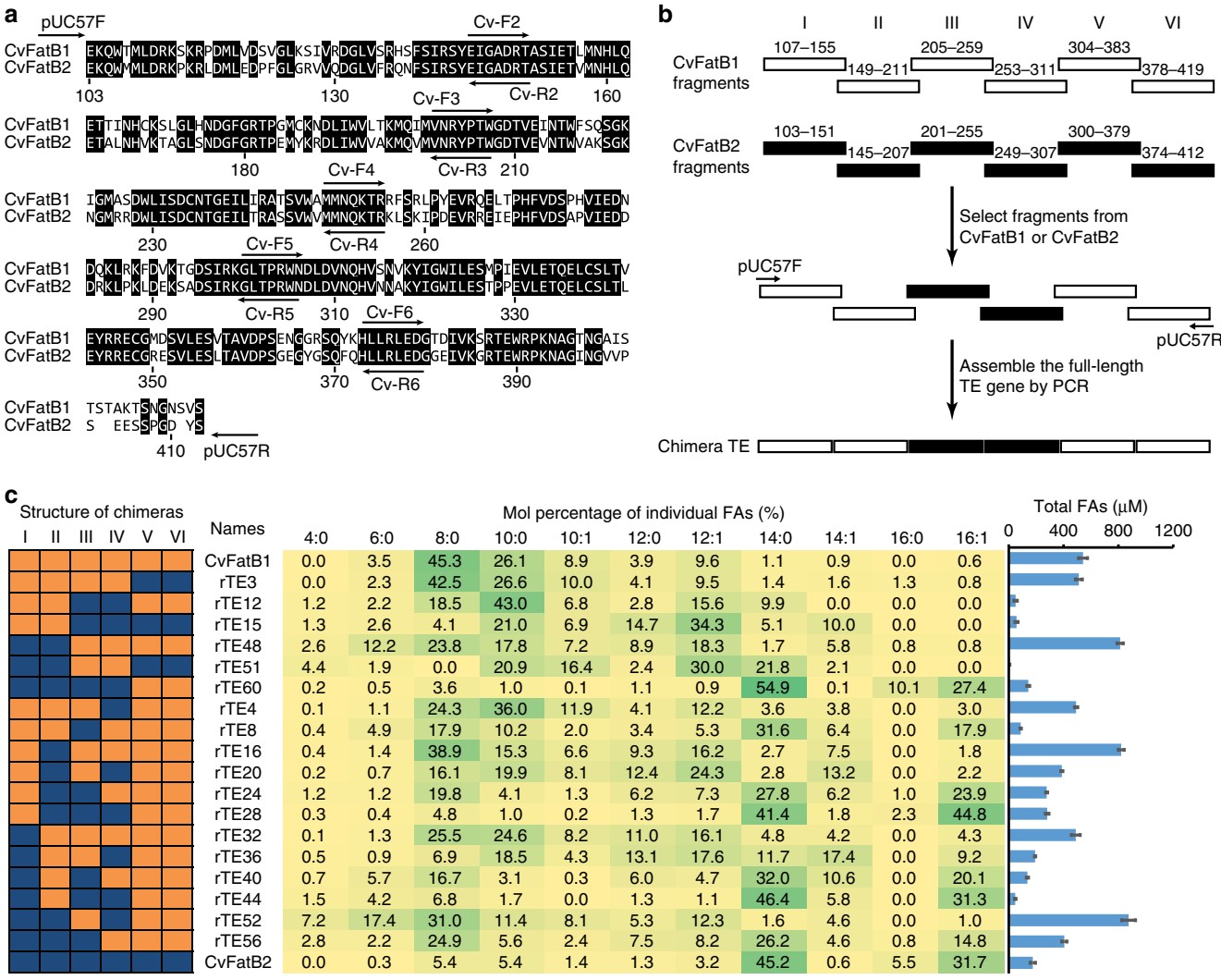

**Fig. 1** Domain-swapping to identify segments that determine substrate specificity. **a** Sequence alignment of CvFatB1 and CvFatB2 identifying identical residues shared between the two enzymes, and the positions of DNA primers used to construct chimeric enzymes; **b** schematic diagram illustrating the construction of chimeric acyl-ACP TEs; **c** substrate specificities of wild-type and chimeric acyl-ACP TEs. The sequence structures of the chimeric acyl-ACP TEs are presented as different color combinations; orange represents the CvFatB1-sourced fragment, and blue represents the CvFatB2-sourced fragment. The central table shows the fatty acid profile generated by the *E. coli* strains expressing each chimeric enzyme, and green shading highlights the major fatty acids produced by each chimera. The graph on the right identifies the total free FAs accumulated in the medium. The data are the average of four replicates, and the error bar represents standard error of the mean

194, 227, 257, and 289) (Supplementary Table 2). Specifically, the first three positions are 100% conserved in all 14 subclass I FatB TEs (i.e., Val, Arg, and Leu, respectively), and position 289 is a Phe residue in all class II/III FatB TEs (Supplementary Table 2). An additional 9 residues can be added to this list if one relaxes the criteria by allowing 15% mismatches in the alignments (i.e., 1 mismatch residue among the 7 subclass II/III FatB TEs sequences, or 2 mismatch residues among the 14 sublass II/III FatB TEs). Specifically, these are at positions 213, 217, 223, 226, 248, and 268, which are conserved among the class I FatB TEs, and at positions 218, 260, and 276 that are conserved among class II/III FatB TEs. An additional two residues at positions 219 and 282 were included in the subsequent experimental analyzes; these were selected because at each position the most frequent residue in each subclass differs from other subclass.

**Site-directed mutagenesis analyses.** Collectively, the above multi-sequence alignment criteria led to the identification of 15

residue differences between CvFatB2 and CvFatB1 that may determine the different substrate specificities of these two TEs (Supplementary Table 2). These hypotheses were experimentally evaluated via site-directed mutagenesis. The point mutants were generated in the CvFatB2 template, primarily substituting the corresponding amino acid from the CvFatB1 sequence at the target position. Moreover, by simultaneously combining mutations at multiple sites we evaluated potential synergy among these residues to affect changes in the substrate specificity of the CvFatB2 enzyme.

The FA profiles generated by *E. coli* cultures expressing the resulting 46 TE mutants were compared to each other and to the two parental enzymes (Fig. 2 and Supplementary Table 3). Cluster analysis of these data categorized the mutant enzymes into two major clades (Fig. 2). One clade of 14 mutants generate predominantly C14/16 FAs, similar to that of the parental CvFatB2 enzyme; these are mutations of residues L257, I260, A276, D282, and L289. These five residues therefore are not significant in determining the different acyl chain length

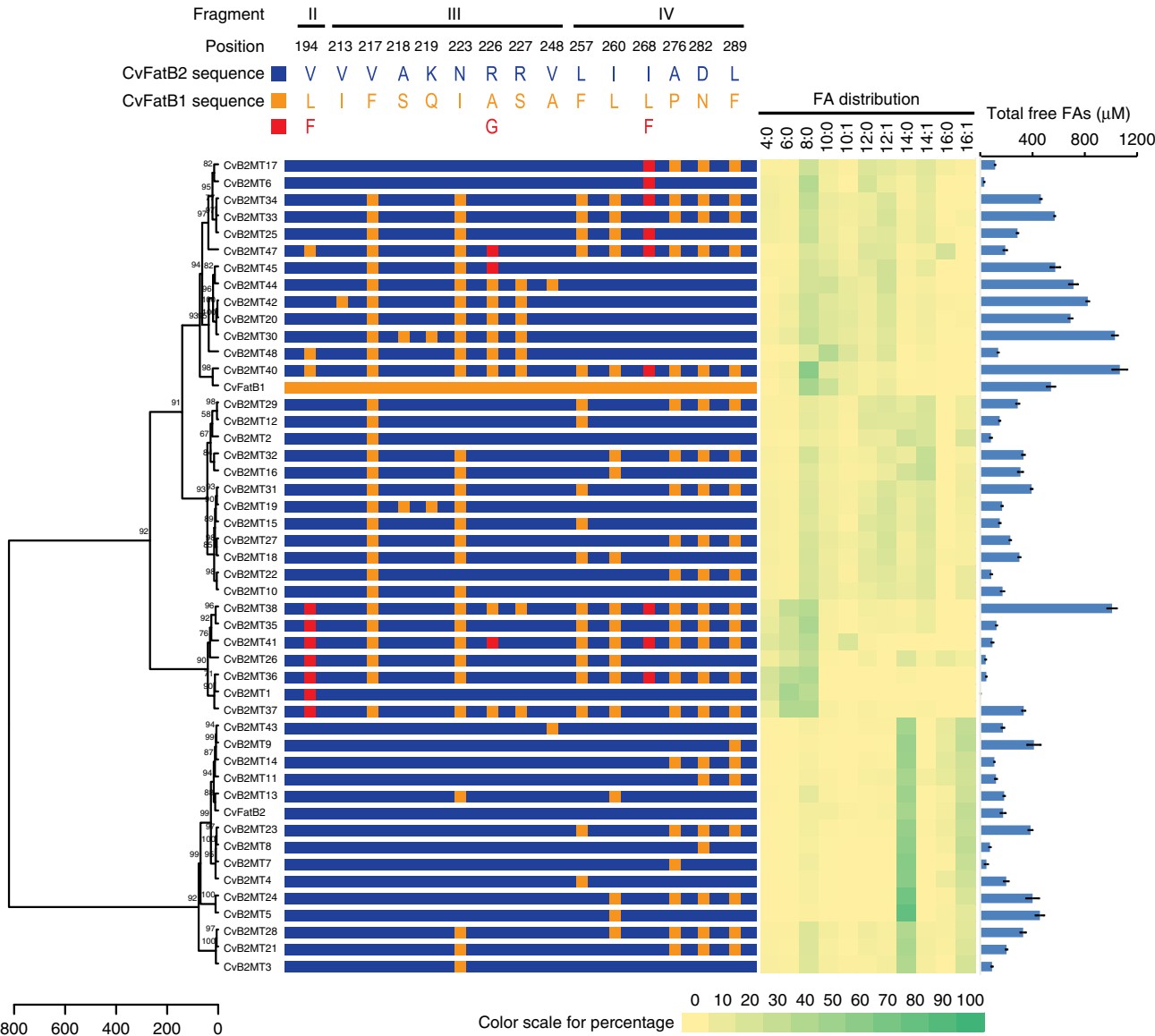

**Fig. 2** Clustering CvFatB2 mutants based on the in vivo FA profiles produced in *E. coli*. The distance matrix was calculated using Euclidean distances, and Ward's method was used to perform agglomerative hierarchical clustering. The approximately unbiased (AU) *p*-values were calculated via multiscale bootstrap resampling with 1000 replicates. The central panel schematically identifies the positions of residues that were mutated in the CvFatB2 sequence (blue colored). The 15 residues that were targeted for mutagenesis are identified and color-coded as either replacement with CvFatB1 residues (orange) or introduction of a residue not found in either CvFatB1 or CvFatB2 (red). The color-coded fatty acid profile is shown on the right side. Major fatty acids produced by each mutant enzyme are highlighted with increasing green coloring (see scale). The graph on the right identifies the total free FAs accumulated in the medium. The data are the average of four replicates, and the error bar represents standard error of the mean

specificity of the TE enzymes. In contrast, the second major clade, containing 32 mutants, generate FA profiles that are similar to that of CvFatB1, producing significantly larger amounts of 4- to 12-carbon FAs, and reduced amounts of C14/16 FAs (Fig. 2 and Supplementary Table 3). These discreet mutations in CvFatB2 therefore change the substrate specificity of the enzyme to be more like CvFatB1, and thereby identifying the small number of residues that are deterministic of the different substrate preferences of the two TEs.

Most significant is mutant CvB2MT40, which carries 11 changes that alter the in vivo FA preference of CvFatB2 to be near identical to that of CvFatB1. All but one of these 11 changes are direct substitution of the CvFatB2 amino acid with the corresponding CvFatB1 amino acid, and the eleventh is the substitution of I168 with the bulkier, aromatic residue Phe. These

data also reveal that mutations located in the domain-swapping fragment III and IV interact to affect changes in the substrate preference of the TE enzyme. Specifically, when only fragment IV-residing residues are changed (as in mutants CvB2MT9, CvB2MT11, CvB2MT20, and CvB2MT23), the substrate preference of CvFatB2 is unaltered, but mutations located in fragment III (i.e., mutants CvB2MT20, CvB2MT30, CvB2MT42, CvB2MT44, CvB2MT45, and CvB2MT48) cluster in a clade that is close to that of CvFatB1-containing clade. But when fragment IV- and fragment III-residing mutants are combined (as in mutants CvB2MT33, CvB2MT34, and CvB2MT40), the substrate preference of the CvFatB2 enzyme becomes most like CvFatB1. These data therefore indicate that fragment IV- and fragment III-residing residues act synergistically to alter the acyl-ACP preference of the CvFatB2 TE. Moreover, the prevalence of

mutations V217F and N223I among the CvFatB2 variants that behave like CvFatB1, indicate that residues V217 and N223 in fragment III are critical in determining the substrate specificity of TE. This latter finding is consistent with the domain-swapping experiments, which indicated the importance of fragment III in determining the substrate specificity of the two enzymes.

Finally, we also tested whether the differences in the FA product profiles produced by each strain might be attributable to different expression level of each variant enzyme. Supplementary Figure 1 shows a representative result of immunological western blot analyzes, of soluble protein extracts prepared from *E. coli* strains expressing CvFatB2, rTE15, rTE28, rTE52, CvB2MT9, CvB2MT20, CvB2MT30, and CvB2MT38. These strains produced FA titers that ranged between 56 μM and 1.03 mM, and these strains also expressed FAs of different acyl chain lengths.

In the case of the chimeric variants (i.e., rTE15, rTE28, and rTE52), these immunological assays are confounded because these proteins present mixed CvFatB1 and CvFatB2 immunological epitopes, therefore they were not further considered in these evaluations. Analysis of the point-mutant variants however could be directly evaluated without this complexity. The accumulation of the CvB2MT9, CvB2MT20, and CvB2MT30 variants was slightly lower than the wild-type CvFatB2 protein, and CvB2MT38 accumulated to even lower levels. However, the FA titers produced with each of these variants were all higher than the CvFatB2 enzyme, with CvB2MT38 producing the highest titer. These results exclude the possibility that the higher FA titers were due to the higher levels of expression by each of the variants. Therefore, the apparent substrate specificity of the acyl-ACP TE, and the productivity of each strain, is primarily attributable to the substrate specificity of the TE enzyme, and catalytic efficiency of the enzyme that is expressed.

In summary, by comparing the FA profiles of this collection of mutants, six residues appear to play a critical role in determining acyl chain length substrate specificity of the TEs: V194 in fragment II; V217, N223, R226, and R227 in fragment III; and I268 in fragment IV. In addition, the effect of these six critical residues is further enhanced, when they are combined with an additional five mutational changes, which reside in fragment IV (L257, I260, A276, D282, and L289), resulting in a variant CvFatB2 enzyme (CvB2MT40) that is most like CvFatB1.

**Synergistic effects between catalysis and selectivity**. The combined domain-swapping and site-directed mutagenesis experiments suggest interactions between domain-swapping fragments III and IV to affect substrate specificity differences between CvFatB1 and CvFatB2. Yet, mutations in domain-swapping fragments IV alone did not affect the substrate specificity of CvFatB2. This is illustrated by considering three specific fragment IV-residing residues at positions 257, 260, and 289 of CvFatB2. These residues were predicted to be important in determining the substrate specificity difference between the two TEs, yet this hypothesis did not stand the site-directed mutagenesis tests described above. We therefore considered the possibility that these three residues fit the selection criteria because they have another important role in catalysis, namely determining the difference in the catalytic activity of the two TE enzymes.

This hypothesis was tested by comparing pairs of mutant enzymes that varied only in mutations in one of the specific sites identified. Thus, for residue L257 we evaluated pairs of mutant enzymes that only differed by the presence or absence of the L257F mutation. Similarly, we compared enzymes that only differed in the presence or absence of the I260L mutation. As illustrated in Table 1, the enzymes that carry the L257F or I260L mutation significantly increased (by up to 3.5-fold) the FA titer of

| Table 1 Pair-wise comparison of total free FAs in the medium produced by *E. coli* strains expressing pairs of TE mutants that differ only by the presence or absence of specific point mutation(s) | | | | |
|---|---|---|---|---|
| **Point mutation(s)** | **Free FAs titer (μM)** | | | |
| | **TEs without the mutation(s)** | | **TEs with the mutation(s)** | |
| L257F | CvB2MT32 | 330.6 | CvB2MT33 | 569.5 |
| | CvB2MT22 | 84.6 | CvB2MT29 | 287.1 |
| | CvB2MT2 | 81.7 | CvB2MT12 | 148.9 |
| | CvB2MT27 | 230 | CvB2MT31 | 393.1 |
| | CvB2MT16 | 308 | CvB2MT18 | 299.9 |
| | CvB2MT10 | 170.9 | CvB2MT15 | 149.6 |
| | CvFatB2 | 175.2 | CvB2MT4 | 199.1 |
| | CvB2MT14 | 108.3 | CvB2MT23 | 384.7 |
| | Paired | *t*-test | *p*-value | 0.012 |
| I260L | CvB2MT31 | 393.1 | CvB2MT33 | 569.5 |
| | CvB2MT27 | 230 | CvB2MT32 | 330.6 |
| | CvB2MT10 | 170.9 | CvB2MT16 | 308 |
| | CvB2MT15 | 149.6 | CvB2MT18 | 299.9 |
| | CvB2MT3 | 90.3 | CvB2MT13 | 183.1 |
| | CvB2MT14 | 108.3 | CvB2MT24 | 400.8 |
| | CvFatB2 | 175.2 | CvB2MT5 | 456.5 |
| | CvB2MT21 | 201.2 | CvB2MT28 | 328.3 |
| | Paired | *t*-test | *p*-value | 0.00022 |
| A276P, D282N, L289F | CvFatB2 | 175.2 | CvB2MT14 | 108.3 |
| | CvB2MT6 | 31.1 | CvB2MT17 | 113.7 |
| | CvB2MT25 | 285.0 | CvB2MT34 | 464.0 |
| | CvB2MT18 | 299.9 | CvB2MT33 | 569.5 |
| | CvB2MT12 | 148.9 | CvB2MT29 | 287.1 |
| | CvB2MT16 | 308.0 | CvB2MT32 | 330.6 |
| | CvB2MT15 | 149.6 | CvB2MT31 | 393.1 |
| | CvB2MT10 | 170.9 | CvB2MT27 | 230.0 |
| | CvB2MT2 | 81.7 | CvB2MT22 | 84.6 |
| | CvB2MT4 | 199.1 | CvB2MT23 | 384.7 |
| | CvB2MT5 | 456.5 | CvB2MT24 | 400.8 |
| | CvB2MT13 | 183.0 | CvB2MT28 | 328.3 |
| | CvB2MT3 | 90.3 | CvB2MT21 | 201.2 |
| | Paired | *t*-test | *p*-value | 0.0025 |

the strain ($p = 0.012$ and $0.00022$ in paired sample *t*-tests, respectively). In a similar, but two-step manner, the significance of the L289F mutation (CvB2MT9) was illustrated in increasing the FA titer of the strain. Specifically, we first determined that the A276P, D282N, and L289F triple mutants increased FA production significantly ($p = 0.0025$ in paired sample *t*-test; Table 1), but when these sites were individually mutated, only the single L289F mutation (CvB2MT9) increased the FA titer by 2-fold, whereas the single-site mutants, A276P (CvB2MT7) and D282N (CvB2MT8), decreased the FA titer of the strain.

We conclude therefore that the reason the three residues, L257, I260, and L289, initially met the theoretical criteria that predicted their roles in determining the substrate preference difference between CvFatB2 and CvFatB1 is due to the fact that these residues meet the same criteria as an explanation for the difference in the catalytic activity between the two enzymes. These data collectively indicate that although the structural features that affect the two catalytic traits (catalytic activity and substrate specificity) can be dissected and can be independently altered and ultimately improved, these trait are not completely independent, but synergistically interact.

**Diverse synthetic acyl-ACP TEs**. The above systematically rationalized strategy to generate and test hypotheses concerning

the structural features that determine the substrate preference of acyl-ACP TEs also generated diverse synthetic enzymes, some of which displayed catalytic traits that are beyond the constraints of the two parental TEs. These catalytic traits are rationalizable in hindsight, based on the substrate specificity model developed from the above experimental data. For example, as illustrated in Fig. 2, the FA profiles produced by mutants CvB2MT1, CvB2MT26, CvB2MT35, CvB2MT36, CvB2MT37, CvB2MT38, and CvB2MT41 define a distinct clade of enzymes that can act on substrates of four-carbon and six-carbon acyl-ACPs, which is a catalytic capability distinct from either CvFatB1 or CvFatB2. This divergent substrate specificity trait is associated with the unique substitution of V194 of CvFatB2 with Phe; substituting V194 with Leu, which is the residue that occurs in CvFatB1, does not have this effect.

Another set of synthetic TE enzymes that display enhanced catalytic trait was generated by the domain-swapping experiments (i.e., rTE48, rTE16, and rTE52), and by the discrete mutants exemplified by variants CvB2MT20, CvB2MT30, CvB2MT38, CvB2MT40, CvB2MT42, and CvB2MT44. The unique character of these variant TE enzymes becomes apparent in the context that CvFatB1 is capable of producing three times more FAs than CvFatB2 when they are expressed in E. coli. Collectively, these data indicate that a combination of mutations at eight residues affect the titer attribute of CvFatB2, these being V213, V217, A218, 219 K, N223, R226, R227, and V248. These nine synthetic enzymes are capable of producing even higher titers of FAs than either parental TE, up to sixfold higher than CvFatB2. These synthetic enzymes therefore have unique attributes that can be used to improve the efficiency for generating advanced biofuels and bioproducts from sugar feedstocks.

**Spatial locations of specificity-determining residues**. The above experimental characterizations identified domains and residues that are critical in determining the difference in the substrate specificity and catalytic activity between CvFatB1 and CvFatB2. Because these structural features are located at distinct positions in the primary structure of the TE proteins, understanding their functionality requires knowledge of the higher-order structure of these enzymes. Although such structural data do not exist for plant-sourced enzymes, crystal structures of several bacterial acyl-ACP TE have been determined, and these have proven useful in

mechanistically understanding plant TEs[13]. Therefore, we modeled the tertiary structure of CvFatB2 using the experimentally determined structure of the acyl-ACP TE from *Lactobacillus plantarum* (2OWN), and evaluated our findings to better understand the structural basis for the acyl chain length specificity and catalytic efficiency of TEs (Fig. 3).

Similar to the bacterial enzyme, the predicted tertiary structure of CvFatB2 contains two hotdog domains linked by a long coil, and is similar to a previously predicted model for the *Arabidopsis* FatB acyl-ACP TE[13]. The structural alignment of the six fragments that were used in the domain-swapped reassembly of chimeric enzymes reveal that the majority of the N-terminal hotdog domain consists of domain-swapping fragments II and III, and the C-terminal hotdog domain consists of fragment V and VI (Fig. 3a). Moreover, fragment III, which we identified as encapsulating half of the causative residues that distinguish the substrate specificity between CvFatB2 and CvFatB1 TEs, encompasses the three β-sheets (β3, β4, and β5) that wrap the central α-helix (α1).

The 2OWN structure reveals a cavity between the central α-helix and the antiparallel β-sheets in the N-terminal hotdog domain (Fig. 4), which may accommodate the acyl chain of the acyl-ACP substrate. Consistent with this model, primarily hydrophobic residues line this cavity, which opens toward the surface where the catalytic triad residues are located[13–16] (Fig. 4a and Supplementary Movie 1).

The modeled CvFatB2 structure is not of sufficient resolution to directly visualize this cavity. We therefore used the structure of 2OWN to identify the cavity-forming residues and mapped these to the primary sequence of CvFatB2. These analyzes defined the residues of CvFatB2 that form the cavity wall, cavity depth, and the cavity opening to the solvent. For example, the 2OWN residues, W62, R97, W116, and Y137, comprise more than 50% of the cavity-wall surface area (Fig. 4b and Supplementary Movie 1), and the corresponding residues in CvFatB2 are W192, R227, W247, and I268. The significance of these residues is revealed by multiple sequence alignment of the 21 experimentally characterized plant acyl-ACP TEs (Supplementary Figure 2). Specifically, W192 is absolutely conserved among all 21 TEs; R227 and W247 are conserved among 19 of the TEs, and are replaced by Ser and Tyr, respectively, in the divergent sequences; similarly, I268 is conserved in 17 of the TEs, and is replaced by hydrophobic residues (e.g., Leu, Val, or Phe) in the 4 divergent TEs. Our site-directed mutagenesis experiments establish the importance of

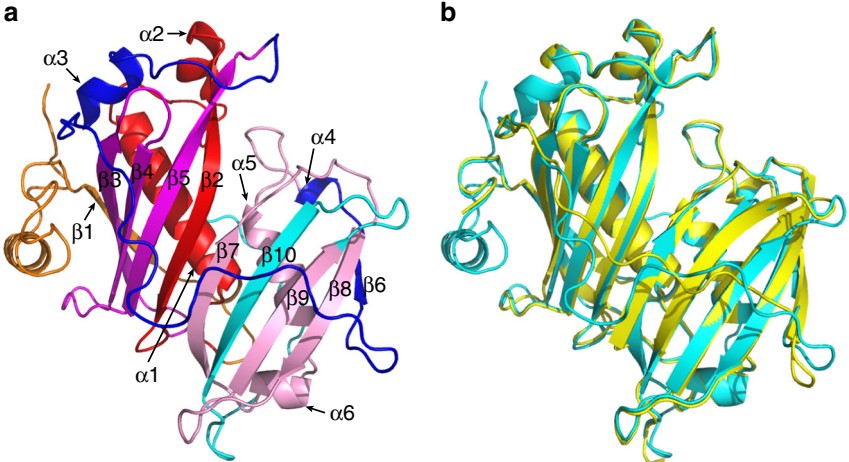

**Fig. 3** Predicted structural model of CvFatB2 compared with the crystal structure 2OWN. **a** Domain-swapping fragments I–VI are shown in orange, red, magenta, blue, pink, and cyan, respectively. The β-sheets are labeled from β1 to β10, and α-helices are labeled from α1 to α6. **b** Structure of CvFatB2 is shown in cyan and the structure of *L. plantarum* acyl-ACP TE, 2OWN, shown in yellow

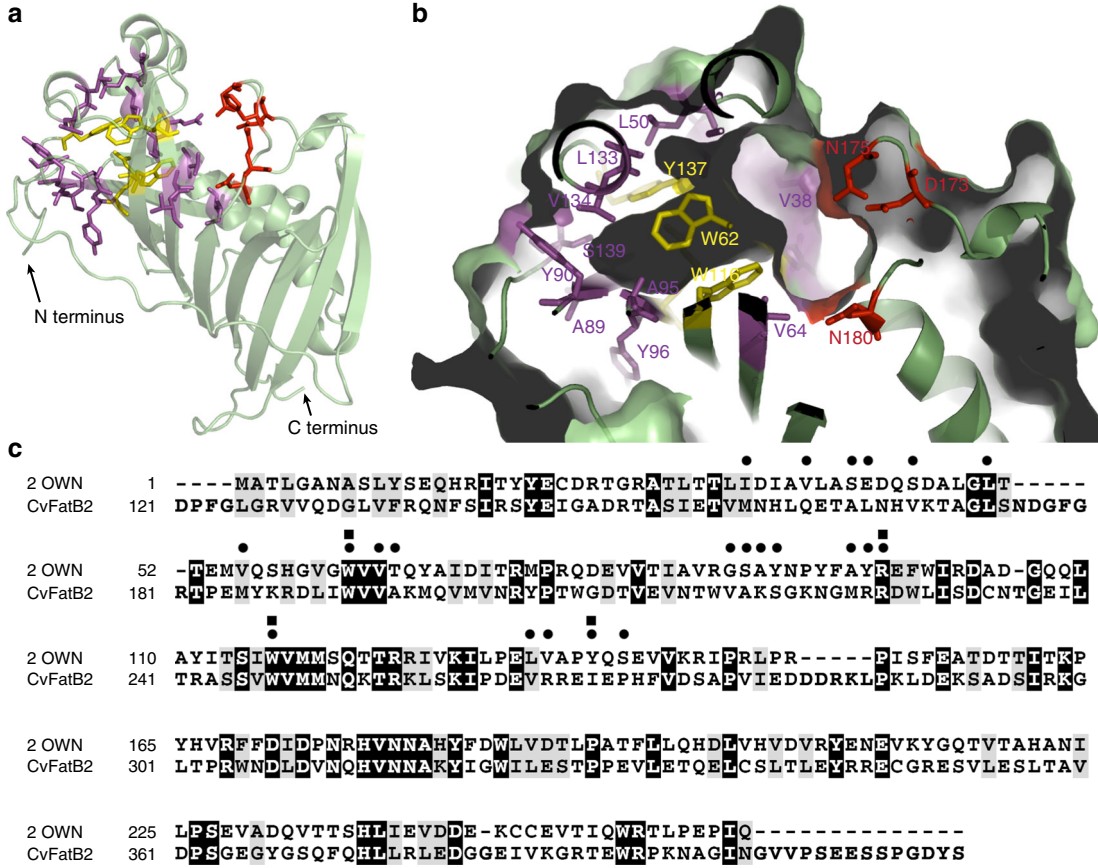

**Fig. 4** Structural analysis of 2OWN and sequence comparison between 2OWN and CvFatB2. **a** Ribbon diagram of the experimentally determined crystal structure of *L. plantarum* acyl-ACP TE, 2OWN. Catalytic residues are shown as red stick models. The residues that form the acyl-binding cavity in the N-terminal hotdog domain are shown as magenta and yellow stick models. The four bulky residues that occupy more than 50% of the surface area of the acyl-binding cavity are shown as yellow stick models. **b** A magnified view of the acyl-binding cavity in the N-terminal hotdog domain. **c** Sequence comparison of 2OWN and CvFatB2; identical residues are black-shaded, and conservative substitutions are gray-shaded. Residues forming the acyl-binding cavity are indicated by dots above the sequences, and the four bulky residues that account for more than 50% of the surface area of this cavity are indicated by squares above the sequences

these residues. Namely, mutations of either R227 or I268 change the substrate specificity of CvFatB2 from primarily preferring C14/16 acyl chains to preferring C8/10 acyl chains. We surmise therefore that these two residues affect the substrate specificity of the TE by determining the shape, size, or chemical nature of the acyl-binding cavity of the enzyme.

Similarly, the seven residues that are part of the segment that forms the ends of two β-sheets connected by a turn structure (i.e., G87, S88, A89, Y90, A95, Y96, and R97) define the bottom, and thus the depth of the acyl-binding cavity of 2OWN (Fig. 4b and Supplementary Movie 1). This region corresponds to residues 217–227 of CvFatB2 (Fig. 4c), and also maps to domain-swapping fragment III. Site-directed mutagenesis within this region experimentally confirmed the importance of four of these residues (V217, N223, R226, and R227) in affecting changes to the substrate preference of CvFatB2. For example, the V217F mutation shifted the substrate specificity of CvFatB2 to preferring the shorter, C8/C10 acyl-ACPs, presumably by decreasing the depth of the acyl-binding cavity.

The V194F mutation in CvFatB2, which is common among six mutant enzymes that display distinct substrate specificity toward 4- to 8-carbon acyl-ACPs (i.e. CvB2MT38, CvB2MT35, CvB2MT41, CvB2MT26, CvB2MT36, and CvB2MT37), maps to residue V64 of 2OWN. In the 2OWN tertiary structure, this

residue is spatially adjacent to residue W116, which is situated on the surface of the acyl-binding cavity. Based on this homology modeling, therefore we surmise that the V194F mutation in CvFatB2 is similarly situated and would have the effect of forcing the cavity-surface residue W116 into the cavity and shrinking the available volume for the substrate, and thereby shifting the substrate specificity of CvFatB2 to shorter acyl chain length substrates.

**Spatial locations of activity-affecting residues**. Three specific mutations in CvFatB2, L257F, I260L, and L289F, appear to affect the catalytic activity of CvFatB2, resulting in *E. coli* strains that produce higher FA titers. The first two of these residues are immersed in a positively charged motif, $R_{255}(R/K)(L/I/F)S(R/K)(I/M/V/L_{260})$, which is conserved among the 21 experimentally characterized acyl-ACP TEs (Supplementary Figure 2). Similarly, a second positively charged conserved motif is positioned adjacent to residue L289 (i.e., $K_{285}(L/V/I)X(K/R)(L/F_{289})$). These two motifs map to two positively charged patches on the surface of the 2OWN structure (i.e., motif 124–129 and motif 143–148), adjacent to the opening of the acyl-binding cavity (Fig. 5). These positively charged surface patches may enable the electrostatic binding of the acidic ACP moiety of the acyl-ACP substrate. Indeed, similar ACP-binding interactions have been implicated

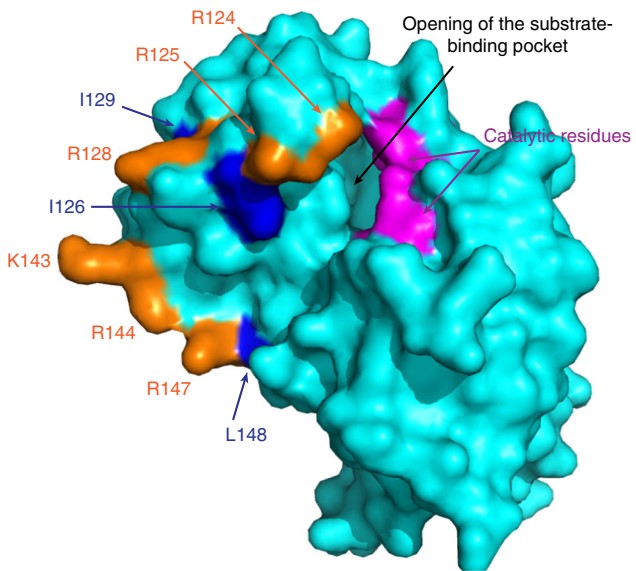

**Fig. 5** Surface of the *L. plantarum* acyl-ACP TE, 2OWN. Positively charged residues in motifs 124–129 and 143–148 are shown in orange. Residues I126, I129, and L148, which correspond to L257, I260, and L289 of CvFatB2, respectively, are shown in blue, and these are adjacent to positively charged motifs. Catalytic residues are shown in magenta

with mutations of FatA1 acyl-ACP TE of *Helianthus annuus* L.[17], and β-ketoacyl-ACP synthase III[18], which also utilize an ACP-bound substrate.

Collectively, therefore, these observations provide a potential explanation of how the L257F, I260L, and L289F mutations increase the activity of the CvFatB2 enzyme. Namely, these mutations change the orientation of surrounding positively charged surface patch, which in turn affects interactions with the acyl-ACP substrates and thereby enhancing catalytic activity. These deductions suggest an explanation of why such near-surface residues (i.e., L257, I260, and L289) have a strong connection with the substrate specificity of the TE enzymes. Namely, that the evolutionary emergence of medium-chain specific acyl-ACP TEs (i.e., subclass II/III) may require the satisfaction of two structural constraints: (1) optimizing the size of the acyl-binding cavity needed to accommodate the size of the medium-chain acyl moiety of the substrate; and (2) the optimization of the interaction between the enzyme and the ACP moiety of the substrate for optimum positioning of the thioester bond that will undergo hydrolysis.

## Discussion

Although there are limited combined set of structural and catalytic data concerning acyl-ACP TEs, the modeling and mutagenesis data presented in this study, combined with earlier studies[17,19–21], build a model that predicates the determinants of substrate specificity. In these experiments, the variant TEs were expressed in *E. coli* strain K27, and the FA titers and FA profiles produced by the resulting strains represented the in vivo activity and substrate specificity of each variant TE enzyme. By comparing the resulting data sets from a large number of variant TEs, we identified important residues that differentiate the selectivity of CvFatB1 from CvFatB2. In CvFatB2, residues V194, V217, N223, R226, R227, and I268 are primarily responsible for the selectivity for C14/16-ACP substrates. The corresponding residues in CvFatB1 (i.e., Leu, Phe, Ile, Ala, Ser, and Leu) are primarily bulkier or more hydrophobic, and are responsible for forming a smaller substrate-binding cavity that better

accommodates the shorter C8/10 acyl chains. Additionally, residues L257, I260, and L289 in CvFatB2, which are located within two positively charged surface patches, are Phe, Leu, and Phe, respectively, in CvFatB1, and these differences optimize the interaction between the TE enzyme and acyl-ACP substrate, and thus enhance the catalytic efficiency for C8/10-ACP substrates.

Mapping the location of these residues that affected either the substrate specificity or catalytic activity of CvFatB2 variants to the three-dimensional model of 2OWN provides a structural understanding of how these mutations affected the catalytic functionality of TEs. These experiments identified two categories of residues that affect the catalytic properties of TEs; those that locate to a cavity and those that map near the surface of the protein. The cavity is the presumed binding pocket of the acyl moiety of the substrate, and its shape and depth would directly affect the acyl chain length specificity of the enzyme. Those residues that define the positive patch near the surface of the protein, adjoining the catalytic residues, presumably bind the negatively charged ACP moiety of the substrate and indirectly affects substrate specificity by increasing the rate of catalysis. Combining these attributes enables synergistic interact and provide an explanation of how 11 amino-acid changes in the CvFatB2 structure, alters the catalytic capability of the enzyme to be like CvFatB1.

The specific substrate-enzyme structural interactions revealed by this study would therefore appropriately present the substrate to the catalytic residues of the enzyme, which are located close to the mouth of the acyl chain substrate-binding site cavity. Such positioning would optimize access by the catalytic residues to the thioester bond that is hydrolyzed in the TE-catalyzed reaction. Analogous conclusions have been drawn from the studies of another class of acyl-ACP-utilizing enzyme, acyl-ACP desaturases, which recognize their substrates via interactions between the acyl moiety of the substrate and the shape and depth of an acyl-binding cavity[22,23]. Characterizations of the interactions between acyl-ACP desaturases with different chain length substrates indicate that the hydrophobic binding energy plays an important role in the substrate selectivity of the desaturases[22]. These interactions in the binding cavity optimize the presentation of the acyl chain to position the appropriate two methylene groups of the substrate next to the di-iron-oxo catalytic cluster that removes hydrogen atoms from the two methylene groups, resulting in the formation of a carbon-carbon double bond[24,25].

We postulate a similar mechanism to explain how acyl-ACP TEs recognize their preferred substrates. The catalytic residues of acyl-ACP TEs are located close to the mouth of the acyl chain substrate-binding site cavity, and these residues will need to be appropriately positioned to initiate the thioester hydrolysis reaction[14,15,26]. The rate of this reaction is determined by the probability of formation of the near-attack conformation between the catalytic residues and the substrate, and the probability of this occurring is dependent upon the free energy of activation[27]. The volume, shape, and hydrophobicity of the cavity can therefore affect the acyl chain-binding energy, which may be utilized by the enzyme to position the substrate at the appropriate near-attack conformation to facilitate catalysis. For a specific acyl-ACP TE, acyl-ACPs of different chain lengths will have different acyl chain-binding energies, which results in different probability to form near-attack conformation and therefore express different reaction rates.

## Methods

**Construction of chimeric acyl-ACP TEs**. Two acyl-ACP TE encoding cDNAs were previously cloned from *C. viscosissima* (CvFatB1 (GenBank Accession JF338906) and CvFatB2 (GenBank Accession JF338907))[10]. The portions of these cDNAs encoding the mature TE peptide (i.e. without the N-terminal chloroplast-

targeting peptide sequence) were codon-optimized for *E. coli* expression, chemically synthesized, and cloned into an expression vector (pUC57) under the transcriptional control of the *lacZ* promoter[10]. Using the primers listed in Supplementary Table 1, six DNA fragments (corresponding to fragments I, II, III, IV, V, and VI) for each TE gene were generated via PCR and used to construct a series of chimeric acyl-ACP TEs.

Fragments I–VI, derived from either CvFatB1 or CvFatB2, were assembled by overlap extension PCR to generate a total of six chimeric acyl-ACP TEs (Fig. 1b). PCR was first performed in a 50-μL reaction mixture containing 10 ng of each fragment, 1× Phusion buffer, 0.2 mM dNTP, and 1 unit of Phusion high-fidelity DNA polymerase (New England Biolabs, USA) using a cycling program of 98 °C for 2 min, 8 cycles of 98 °C for 10 s, 50 °C for 15 s, and 72 °C for 20 s, and a final extension step of 72 °C for 5 min. Upon completion of this PCR program, 0.5 μM pUC57F and pUC57R primers were immediately added into the reaction mixture, followed by a second PCR cycling program of 98 °C for 2 min, 30 cycles of 98 °C for 10 s, 54 °C for 15 s, and 72 °C for 20 s, and a final extension step of 72 °C for 5 min. The targeted full-length gene products were separated by electrophoresis in a 1% agarose gel, recovered from the gel using the QiaQuick gel extraction kit (Qiagen, Valencia, CA, USA), and cloned into the pUC57 vector using the *Bam*HI and *Eco*RI restriction sites. The chimeric TEs were confirmed by sequencing both strands of all constructs.

**Structural modeling of CvFatB2.** Primary protein sequences were subjected to tertiary structure modeling with I-TASSER (http://zhanglab.ccmb.med.umich.edu/I-TASSER/), using experimentally determined crystal structure of acyl-ACP TE from *L. plantarum* (2OWN) as a template[28]. The resulting tertiary structures were viewed and analyzed with the PyMOL Molecular Graphics System, Version 1.5.0.4 (Schrödinger, LLC).

**Site-directed mutagenesis.** Point mutations were generated in CvFatB2 using the QuikChange II site-directed mutagenesis kit (Agilent Technologies, USA), according to the procedures provided by the manufacturer. Multiple mutations were generated sequentially. The authenticity of all site-directed mutants was confirmed by sequencing both strands of the plasmids.

**In vivo activity of acyl-ACP TE variants.** Chimeric acyl-ACP TEs and site-directed mutants were expressed in *E. coli* strain K27 (*E. coli* Genetic Stock Center at Yale, CGSC# 5478) (*fadD88*), which contains a mutation in the *fadD* gene that disrupts β-oxidation and results in the accumulation of free FAs in the growth medium. Free FAs that accumulated in the medium were extracted and analyzed by gas chromatography-mass spectrometry[10]. For each TE construct, four individual colony isolates were cultured in 2 mL Luria-Bertani (LB) medium supplemented with 100 mg L$^{-1}$ carbenicillin. When the culture reached an OD600 of ~0.7, the growth medium was replaced with 3 mL of M9 minimal medium (47.7 mM Na$_2$HPO$_4$, 22.1 mM KH$_2$PO$_4$, 8.6 mM NaCl, 18.7 mM NH$_4$Cl, 2 mM MgSO$_4$, and 0.1 mM CaCl$_2$) supplemented with 0.4% glucose and 100 mg L$^{-1}$ carbenicillin. Acyl-ACP TE expression was induced by adding 0.4 mM isopropyl-β-D-thiogalactopyranoside. After 40 h cultivation at 30 °C, the cultures showed very similar cell density, and free FAs were extracted from the medium and analyzed[10]. FA productivity for each acyl-ACP TE was measured by subtracting the concentration of FAs produced by *E. coli* expressing a control plasmid (pUC57) that lacks the TE gene. The FA titer is used as a best guide of TE activity in *E. coli*. The mol% of individual FAs was calculated as a quantification of the substrate specificity of each acyl-ACP TE.

**Determination of acyl-ACP TE protein expression level.** The expression levels of the chimeric and mutant acyl-ACP TE proteins were quantified by extracting soluble proteins from cell pellets and subsequently performing immunoblot analyzes. Cells were collected after 40 h of cultivation, suspended, and incubated for 30 min in lysis buffer (50 mM Tris-HCl (pH 8.0), 150 mM NaCl, 10% glycerol, 0.6 mM phenylmethanesulfonylfluoride, and 0.2 mg mL$^{-1}$ lysozyme). Following sonication for 5 s, the suspension was subjected to centrifugation at 13 000 × *g* for 5 min, and the soluble protein fraction was recovered as the supernatant. Protein concentration was quantified using the Bio-Rad Protein Assay Kit (Bio-Rad, USA). Aliquots of the protein extract, each containing 35 μg of total soluble proteins, were separated via SDS-polyacrylamide gel electrophoresis in a 12% polyacrylamide gel and the proteins were transferred to nitrocellulose membrane. Membranes were reacted with the primary CvFatB2 antibody (made by Hybridoma Facility at Iowa State University) at 1:2000 dilution, and the secondary antibody goat-anti-mouse IgG (H + L)-horseradish peroxidase (Catalog Number 1706516, Bio-Rad) at 1:3000 dilution. Immuno-detection was performed using an enhanced chemi-fluorescence western blotting detection kit according to the manufacturer's instructions (Thermo Scientific, USA) and visualized using a ChemiDoc™ XRS+ System (Bio-Rad). Images were analyzed with Image Lab™ Software (Bio-Rad).

**Data availability.** The data sets generated and/or analyzed during the current study are available from the corresponding author on reasonable request.

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

## Acknowledgements

This work was supported by the U.S. National Science Foundation through its Engineering Research Center Program (Award No. EEC-0813570), leading to the Center for Biorenewable Chemicals (CBiRC), headquartered at Iowa State University and including Rice University, the University of California, Irvine, the University of New Mexico, the University of Virginia, and the University of Wisconsin-Madison. The authors thank Drs. M. Ann D.N. Perera and Zhihong Song of the W.M. Keck Metabolomics Research Laboratory at Iowa State University for assistance with fatty acid analysis.

## Author contributions

F.J. conducted bioinformatical analysis and generated acyl-ACP TE variants. F.J. and L.Z. measured in vivo activities and substrate specificities of those acyl-ACP TE variants in *E. coli*. M.D.Y.-N. and B.J.N. supervised this study. F.J., M.D.Y.-N. and B.J.N. analyzed the data and wrote the manuscript. All authors read and approved the final manuscript.

## Additional information

**Competing interests:** The authors declare no competing interests.

