## [Peer Review File · Nature Communications]

Reviewers' comments:

Reviewer #1 (Remarks to the Author):

Jing et. al., investigated the substrate specificity of acyl-ACP thioesterases by combining domain swapping (which the authors refer to as domain shuffling), site-directed mutagenesis, bioinformatic analyses and structural modelling approaches. The authors concluded the determinants for TEs' substrate specificity are based on the nature of the acyl-chain-binding-pocket-forming residues along with residues that reside in the predicted TE-ACP interaction interface.

The authors were able to dissect and pinpoint residues that affect the substrate specificity by comparing two highly similar TEs that differ in substrate preference, laying the foundation for understanding the specificity of the enzyme. In addition, the creation of variants with new intermediate specificities was a very nice outcome of the study that has potential biotechnological application.

This is a nice paper at its core which represents a long overdue analysis of specificity for plant thioesterases.

The experiments were logically designed and well executed. One strength of the paper is that the results are creatively presented in a manner that is easy to visualize.

On the other hand images of the structural model could benefit from applying labels to residues manually instead of using the Pymol autolabeling function. The way of showing the structure model could use some improvement. For example, in Figure 4 panels A and B, the labeling of the residue should be added manually instead of using the Pymol defaults. It also will be nicer if the authors can comparatively show how these corresponding residues are mapped onto the model of CvFatB2 in Figure4 panel B, and Figure 5.

Use of the word shuffling is misleading and usually reserved for approaches in which random fragment assemblies are created. In this case, fragments were intentionally swapped to make a small sample of 18 predefined patterns, representing only a small fraction of the possible 718 combinations.

While the writing is generally very clear and concise, it is often too general.

For instance, the title and abstract offer little information other than generalities. For example a title such as Two distinct domains contribute to the substrate chain length selectivity of plant acyl-ACP thioesterase would be more informative. (I am sure with more thought, more informative titles could be proposed). Similarly the abstract lacks specific findings from the study.

If two enzymes with different specificity are being discussed, it should be spelled out that CvFatB1 is primarily active on C-8:0 and C-10:0 substrates whereas CvFatB2 is primarily active on C-14:0 and C-16:1.

An attempt at defining the domains by amino acid groups or in the region of x-y in the sequence should be used to define the regions in the abstract that define the cavity and the patch.

A section close to the beginning of the introduction should be devoted to explaining the specificity of CvFatB1 and CvFatB2, because the naming appears to be so confusing that even the authors can't get it right. On line 89 they state: "Specifically we utilized two FatB type acyl-ACP TEs from *Cuphea viscosissima* (CvFatB1 and CvFatB2) that share more than 70% sequence identity, but they functionally belong to Subclass I and Subclass III FatB TEs, respectively" If I am correct, these represent Subclass III and I, respectively. At any rate this needs to be crystal clear in the abstract and introduction or the reader will not be able to follow the paper.

While the data regarding thioesterase specificity is quantitative and convincing, data regarding activity is much less so. Figure S3 shows a range of expression of the TE variants in *E. coli*. It is not clear if these levels of accumulation represent soluble or insoluble TE. Was total extract subject to western analysis or the soluble phase after clarification by centrifugation? Also titers are presented to four and sometimes five significant figures in Figure S2, which implies precision that is simply absent from this in vitro analytical method. Unless the samples were all induced at precisely the same OD and harvested at identical final ODs, with exactly equivalent TE

accumulation in vivo, these numbers are a guide at best. This should be reflected in the text. There are a few minor typos, for example:

1. Page 4, line 102 "TEs" should be "TE"; line 106 delete the first "and"
2. Page 20, line 457, "Ile" should be "Leu"

Reviewer #2 (Remarks to the Author):

This study has attempted to define, at the amino acid level, the sequence determinants for different subclasses of higher plant acyl-ACP thioesterases. On one level, these kind of projects initially seem deceptively straightforward - two different enzyme activities generated by two closely related polypeptide sequences. However, once embarked on, they often involve a brutal amount of work and fail to fully resolve the issue. I think that this is the situation here, where the authors have established a very effective assay with which to test chimeras and SDMs for altered thioesterase activity, but ultimately, the definitive identification of critical determinant residues is lacking, at least to the point where it is possible to say if residues A, B and C are varied, then substrate-specificity is switched from N to Y. Of course, that is quite a big ask, and in some cases, differences in substrate specificity are quite likely to be determined by multiple residues. But in this manuscript, I struggled to follow the narrative that the authors presumably wanted to tell, and some of the experiments didn't seem logical. For example, it wasn't clear to me why the authors defined 6 domains for their chimeras, but then effectively only tested the first 4. I mean, the assumption seems to have been that because V and VI didn't directly alter the substrate-specificity of the TE, then these two regions could be ignored. But then subsequently, when thinking about the homology modelled structure against the bacterial TE, then it is obvious that different regions which are not linear can be in proximity in 3D. One is also left wondering why the authors didn't start with a homology-modelled structure and use that to inform their chimeras and/or SDM. Some other points.

> The so-called new activities against C4-6 substrates could potentially be interesting, but the actual activity of these enzymes seems to be almost negligible, at least in the assay used. Of course, it could be that the system used for the in vivo TE assay is biased against certain acyl chain lengths.

> I found the manuscript hard to follow, especially the section on SD mutants described in Fig 2. In Fig 1 and associated text, domain III was identified as being the major determinant of differences in TE activity. However, nearly half of the residues selected for point mutation in the next section are not in domain III, but adjacent domains. Of course, their choice can be seen as logical when it is presented in the context of sequence analysis of TEs from other species, but again, one is left wondering, what order was all of this work done in, and wouldn't it have been better to start with a detailed in silico analysis before embarking on making chimeras and SDMs?

Reviewer #3 (Remarks to the Author):

The manuscript by Jing et al. describes the structural determinants of plant acyl-ACP thioesterases of the FatB type for the chain-length specificity of the acyl-ACP substrate, employing the two enzymes from *Cuphea viscosissima* as models. Acyl-ACP thioesterases are crucial enzymes of fatty acid de novo synthesis in the plastids of plants involved in determining the length of the acyl chain that is exported from the plastid. They are thus important for determining the quality of plant oils, in particular those with short or medium chain fatty acids. In their manuscript, Jing et al. expressed different protein mutants of FatB1 or FatB2 from *C. viscosissima* (domain shuffling and point mutations) in *E. coli*. Measurements of fatty acids released from *E. coli* were taken as a measure of the overall activity and substrate specificity of the mutant proteins. After modeling the

structure based on the *Lactobacillus* thioesterase 20WN, the authors could explain most of the effects of amino acid exchanges on the activity and specificity.

The author should include in their discussion that measurements of fatty acids released from transformed *E. coli* reflect the specificity of the enzymes in *E. coli*, because the fatty acid synthesis machinery and the fatty acid composition are presumably not the same as in plants. This might be relevant when considering to transfer the results to transgenic plants.

Can the authors please comment on the question why the *Lactobacillus* structure 20WN and not the *Pseudomonas* structure 1BVQ (Mayer et al., 2005) was used as template for the prediction? Is the sequence identity of CvFatB2 with the *Lactobacillus* sequence higher?

Page 3, line 63: "The biotechnological focus on acyl-ACP TEs was primed by the discovery that this enzyme is the single determinant that enables seeds of certain plants to produce laurate-rich oils."

Presumably, other genes in addition to FatB are also involved in the accumulation of high amounts of 12:00 in seed oil, e.g. KAS and enzymes of TAG synthesis.

Suppl. Fig. S3:

Why do we see multiple bands in the Western blot? Does the top band or the band in the middle of CvFatB2 represent the thioesterase protein? Are the sizes of the bands in CvFATB2 rTE15 similar?

Page 14, line 320, "Namely, despite the wide difference in fatty acid titers, the expressed level of the TE was similar for CvFatB2 and the CvB2MT9, CvB2MT20, and CvB2MT30 variants. The single outlier in these analyses was CvMT38."

To me it seems that expression in CvB2MT9, CvB2MT20, and CvB2MT30 is a little lower, and in CvMT38 it is quite lower compared to CvFatB2.

Page 16, line 386: "CvB2MT20, CvB2MT30, CvB2MT38, CvB2MT30, CvB2MT40, CvB2MT42 and CvB2MT44 "

Mutant CvB2MT30 is mentioned two times.

Page 17, line 387: "The unique character of these variant TE enzymes becomes apparent in the context that CvFatB1 is capable of producing 3 times more fatty acids than CvFatB2 when similarly expressed in *E. coli*."

Is the protein accumulation of FatB1 and FatB2 in *E. coli* comparable?

Page 18, line 417: "Similar to the bacterial enzyme, the predicted tertiary structure of CvFatB2 contains two hotdog domains linked by a long coil, and is similar to a previously predicted model for the *Arabidopsis* acyl-ACP TE"

Please change "*Arabidopsis* acyl-ACP TE" to "*Arabidopsis* acyl-ACP FatB"

Page 26: Are references 14 and 15 identical?

Responses to Reviewers' comments

Reviewer #1 (Remarks to the Author):

Jing et. al., investigated the substrate specificity of acyl-ACP thioesterases by combining domain swapping (which the authors refer to as domain shuffling), site-directed mutagenesis, bioinformatic analyses and structural modelling approaches. The authors concluded the determinants for TEs' substrate specificity are based on the nature of the acyl-chain-binding-pocket-forming residues along with residues that reside in the predicted TE-ACP interaction interface.

The authors were able to dissect and pinpoint residues that affect the substrate specificity by comparing two highly similar TEs that differ in substrate preference, laying the foundation for understanding the specificity of the enzyme. In addition, the creation of variants with new intermediate specificities was a very nice outcome of the study that has potential biotechnological application.

This is a nice paper at its core which represents a long overdue analysis of specificity for plant thioesterases.

The experiments were logically designed and well executed. One strength of the paper is that the results are creatively presented in a manner that is easy to visualize.

On the other hand images of the structural model could benefit from applying labels to residues manually instead of using the Pymol autolabeling function. The way of showing the structure model could use some improvement. For example, in Figure 4 panels A and B, the labeling of the residue should be added manually instead of using the Pymol defaults. It also will be nicer if the authors can comparatively show how these corresponding residues are mapped onto the model of CvFatB2 in Figure4 panel B, and Figure 5.

AUTHORS' RESPONSE: Figures 4 and 5 have been modified as the reviewer's suggestion of manually labeling the specified residues, rather than the Pymol autolabeling.

Use of the word shuffling is misleading and usually reserved for approaches in which random fragment assemblies are created. In this case, fragments were intentionally swapped to make a small sample of 18 predefined patterns, representing only a small fraction of the possible 718 combinations.

AUTHORS' RESPONSE: We have changed the word "shuffling" to "swapping"

While the writing is generally very clear and concise, it is often too general.

For instance, the title and abstract offer little information other than generalities. For example a title such as Two distinct domains contribute to the substrate chain length selectivity of plant acyl-APC thioesterase would be more informative. (I am sure with more thought, more

informative titles could be proposed). Similarly the abstract lacks specific findings from the study.

If two enzymes with different specificity are being discussed, it should be spelled out that CvFatB1 is primarily active on C-8:0 and C-10:0 substrates whereas CvFatB2 is primarily active on C-14:0 and C-16:1.

An attempt at defining the domains by amino acid groups or in the region of x-y in the sequence should be used to define the regions in the abstract that define the cavity and the patch.

AUTHORS' RESPONSE: We changed to a more informative title as the reviewer suggested; specifically, "Two distinct domains contribute to the substrate acyl chain length selectivity of plant acyl-APC thioesterase"

A section close to the beginning of the introduction should be devoted to explaining the specificity of CvFatB1 and CvFatB2, because the naming appears to be so confusing that even the authors can't get it right. On line 89 they state: "Specifically we utilized two FatB type acyl-ACP TEs from *Cuphea viscosissima* (CvFatB1 and CvFatB2) that share more than 70% sequence identity, but they functionally belong to Subclass I and Subclass III FatB TEs, respectively" If I am correct, these represent Subclass III and I, respectively. At any rate this needs to be crystal clear in the abstract and introduction or the reader will not be able to follow the paper.

AUTHORS' RESPONSE: We have made the suggested change and correction; see page 4.

While the data regarding thioesterase specificity is quantitative and convincing, data regarding activity is much less so. Figure S3 shows a range of expression of the TE variants in *E. coli*. It is not clear if these levels of accumulation represent soluble or insoluble TE. Was total extract subject to western analysis or the soluble phase after clarification by centrifugation?

AUTHORS' RESPONSE: We have added text to the Methods section (last paragraph of page 16) and to the figure legend (Figure S3 which is now renamed as Supplementary Figure 1) to indicate that the western analysis was conducted with proteins recovered in the soluble phase after clarification by centrifugation.

Also titers are presented to four and sometimes five significant figures in Figure S2, which implies precision that is simply absent from this *in vitro* analytical method.

AUTHORS' RESPONSE: We have made the corrections in Figure S2 concerning the significant figures. Figure S2 is now changed to Supplementary Table 3 in the supplementary data file.

Unless the samples were all induced at precisely the same OD and harvested at identical final ODs, with exactly equivalent TE accumulation in vivo, these numbers are a guide at best. This should be reflected in the text.

AUTHORS' RESPONSE: All cultures were induced at the start of the inoculation and cells were harvested at the same time. These cultures showed a very similar growth rate, and cursory examination of SDS-PAGE analysis indicated that TEs were expressed at similar levels, which was also confirmed by immunological experiments shown in Figure S3 (now Supplementary Figure 1). Therefore, we agree with the reviewer that the FA titer is just a best guide of enzyme activity, and have appropriately added text in the Method section (second paragraph of page 16).

There are a few minor typos, for example:

1. Page 4, line 102 “TEs” should be “TE”; line 106 delete the first “and”
2. Page 20, line 457, “Ile” should be “Leu”

AUTHORS' RESPONSE: All the typos have been corrected

Reviewer #2 (Remarks to the Author):

This study has attempted to define, at the amino acid level, the sequence determinants for different subclasses of higher plant acyl-ACP thioesterases. On one level, these kind of projects initially seem deceptively straightforward - two different enzyme activities generated by two closely related polypeptide sequences. However, once embarked on, they often involve a brutal amount of work and fail to fully resolve the issue. I think that this is the situation here, where the authors have established a very effective assay with which to test chimeras and SDMs for altered thioesterase activity, but ultimately, the definitive identification of critical determinant residues is lacking, at least to the point where it is possible to say if residues A, B and C are varied, then substrate-specificity is switched from N to Y. Of course, that is quite a big ask, and in some cases, differences in substrate specificity are quite likely to be determined by multiple residues.

AUTHORS' RESPONSE: We do not quite understand the reviewer's comments, because our data clearly identify combination of residues that when switched results in altering the substrate specificity of one thioesterase to the substrate specificity of another, and this was achieved by careful consideration of sequence differences modelled on a structure that is an extrapolation of a known crystal structure. Indeed, both of the other reviewers have come to this conclusion. Moreover, the data we present are consistent with a structural model of the two enzymes, and are consistent with general principles of fatty acyl-ACP desaturases,

whose substrate specificities are determined by the shape and size of the substrate binding cavity; and these comparisons are made in the Discussion of the current paper.

But in this manuscript, I struggled to follow the narrative that the authors presumably wanted to tell, and some of the experiments didn't seem logical. For example, it wasn't clear to me why the authors defined 6 domains for their chimeras, but then effectively only tested the first 4. I mean, the assumption seems to have been that because V and VI didn't directly alter the substrate-specificity of the TE, then these two regions could be ignored. But then subsequently, when thinking about the homology modelled structure against the bacterial TE, then it is obvious that different regions which are not linear can be in proximity in 3D. One is also left wondering why the authors didn't start with a homology-modelled structure and use that to inform their chimeras and/or SDM.

AUTHORS' RESPONSE: In fact we conducted homology modeling in parallel with the domain swapping experiments, which enabled us to identify the boundaries between the fragments for the domain swapping experiments. These domain swapping experiments enabled us to narrow the experimental search of substrate determining residue differences between the two enzymes. Thus the domain swapping experiments indicated that the different residues in the C-terminal 2 fragments are not important in determining substrate specificities of the 2 enzymes. Hence enabling us to focus on the first 4 Fragments

Some other points.

> The so-called new activities against C4-6 substrates could potentially be interesting, but the actual activity of these enzymes seems to be almost negligible, at least in the assay used. Of course, it could be that the system used for the in vivo TE assay is biased against certain acyl chains lengths.

AUTHORS' RESPONSE: Although the activities of some of these novel enzymes are relatively low (e.g., CvB2MT26, CvB2MT35, CvB2MT36, and CvB2MT41), two specific variants (CvB2MT37 and CvB2MT38) show even higher activities than the parental CvFatB2 enzyme (see Figure 2 and Supplementary Table 3). Especially CvB2MT38 produced 1009 μ M total FAs, of which 8.5 % is C4 FA and 31% is C6 FA.

> I found the manuscript hard to follow, especially the section on SD mutants described in Fig 2. In Fig 1 and associated text, domain III was identified as being the major determinant of differences in TE activity. However, nearly half of the residues selected for point mutation in the next section are not in domain III, but adjacent domains. Of course, their choice can be seen as logical when it is presented in the context of sequence analysis of TEs from other species, but again, one is left wondering, what order was all of this work done in, and wouldn't it have been better to start with a detailed in silico analysis before embarking on making chimeras and SDMs?

AUTHORS' RESPONSE: Many of the experiments described were conducted in parallel. However, we constructed the manuscript in an order that provides a logical explanation of the data. And indeed, *in silico* modeling was used to guide the domain swapping experiments and the site directed mutagenesis studies.

Reviewer #3 (Remarks to the Author):

The manuscript by Jing et al. describes the structural determinants of plant acyl-ACP thioesterases of the FatB type for the chain-length specificity of the acyl-ACP substrate, employing the two enzymes from *Cuphea viscosissima* as models. Acyl-ACP thioesterases are crucial enzymes of fatty acid de novo synthesis in the plastids of plants involved in determining the length of the acyl chain that is exported from the plastid. They are thus important for determining the quality of plant oils, in particular those with short or medium chain fatty acids. In their manuscript, Jing et al. expressed different protein mutants of FatB1 or FatB2 from *C. viscosissima* (domain shuffling and point mutations) in *E. coli*. Measurements of fatty acids released from *E. coli* were taken as a measure of the overall activity and substrate specificity of the mutant proteins. After modeling the structure based on the *Lactobacillus* thioesterase 20WN, the authors could explain most of the effects of amino acid exchanges on the activity and specificity.

The author should include in their discussion that measurements of fatty acids released from transformed *E. coli* reflect the specificity of the enzymes in *E. coli*, because the fatty acid synthesis machinery and the fatty acid composition are presumably not the same as in plants. This might be relevant when considering to transfer the results to transgenic plants.

AUTHORS' RESPONSE: As suggested by the reviewer, we have added additional text in the first paragraph of Discussion on page 13, indicating that the *in vivo* generated fatty acids reflect the specificity of the enzymes in transformed *E. coli* cells.

Can the authors please comment on the question why the *Lactobacillus* structure 20WN and not the *Pseudomonas* structure 1BVQ (Mayer et al., 2005) was used as template for the prediction? Is the sequence identity of CvFatB2 with the *Lactobacillus* sequence higher?

AUTHORS' RESPONSE: Although both 20WN and 1BVQ share very similar sequence identity with CvFatB2 (25% and 23%, respectively), the rationale for using the 20WN structure as the template was based on the fact that its catalytic function is acyl-ACP thioesterase, identical to CvFatB2, whereas 1BVQ catalyzes a homologous reaction, the hydrolysis of 4-hydroxybenzoyl-CoA. It should be noted that the earlier study cited by the

reviewer (Mayer et al., 2005) was conducted at a time when the structure of 2OWN was not available in the database, it was deposited in 2007.

Page 3, line 63: "The biotechnological focus on acyl-ACP TEs was primed by the discovery that this enzyme is the single determinant that enables seeds of certain plants to produce laurate-rich oils."

Presumably, other genes in addition to FatB are also involved in the accumulation of high amounts of 12:00 in seed oil, e.g. KAS and enzymes of TAG synthesis.

AUTHORS' RESPONSE: The reviewer's comment is correct; KAS enzymes affect the pool size of different acyl-ACPs and therefore influence the fatty acids profiles. We have therefore changed the word "single" to "major" on page 3.

Suppl. Fig. S3:

Why do we see multiple bands in the Western blot? Does the top band or the band in the middle of CvFatB2 represent the thioesterase protein? Are the sizes of the bands in CvFATB2 rTE15 similar?

AUTHORS' RESPONSE: The multiple bands are probably the result of post-extraction proteolytic clipping of the expressed thioesterases. Because "all" the bands are immunologically recognized by our CvFatB2 antibody, we have integrated the signal from the entire collection of bands to compare the relative expression of the thioesterase variants. We have added appropriate text in the figure legend. Figure S3 is now Supplementary Figure 1.

Page 14, line 320, "Namely, despite the wide difference in fatty acid titers, the expressed level of the TE was similar for CvFatB2 and the CvB2MT9, CvB2MT20, and CvB2MT30 variants. The single outlier in these analyses was CvMT38."

To me it seems that expression in CvB2MT9, CvB2MT20, and CvB2MT30 is a little lower, and in CvMT38 it is quite lower compared to CvFatB2.

AUTHORS' RESPONSE: The reviewer is correct, and we have altered the text appropriately. Please see the second paragraph of page 8. Page number changed because the Method section and figures were moved to the end of the manuscript.

Page 16, line 386: "CvB2MT20, CvB2MT30, CvB2MT38, CvB2MT30, CvB2MT40, CvB2MT42 and CvB2MT44 "

Mutant CvB2MT30 is mentioned two times.

AUTHORS' RESPONSE: We thank the reviewer for catching our error. It has been corrected. See page 10.

Page 17, line 387: "The unique character of these variant TE enzymes becomes apparent in the context that CvFatB1 is capable of producing 3 times more fatty acids than CvFatB2 when similarly expressed in *E. coli*."

Is the protein accumulation of FatB1 and FatB2 in *E. coli* comparable?

AUTHORS' RESPONSE: We did not directly compare the protein accumulation of CvFatB1 and CvFatB2 in *E. coli*. But based on our western blot analysis on some TE point mutants, we have concluded that the fatty acid production of each strain is primarily attributable to the enzyme activity of expressed TE. We have rephrased this sentence to be more accurate. See page 10.

Page 18, line 417: "Similar to the bacterial enzyme, the predicted tertiary structure of CvFatB2 contains two hotdog domains linked by a long coil, and is similar to a previously predicted model for the Arabidopsis acyl-ACP TE"

Please change "Arabidopsis acyl-ACP TE" to "Arabidopsis acyl-ACP FatB"

AUTHORS' RESPONSE: We have made the suggested change.

Page 26: Are references 14 and 15 identical?

AUTHORS' RESPONSE: Again, we thank the reviewer for catching our error. We have made the correction.

REVIEWERS' COMMENTS:

Reviewer #1 (Remarks to the Author):

The manuscript is much improved, but I am still frustrated that the abstract and discussion allude to details that have to be dug out of the results section, often via reference to figures or tables. The abstract could be significantly strengthened to include some detail of the domains. perhaps the domains should be named and those n

Also authors refer to "Bioengineering of the two domains leads to predictable changes in substrate specificity, many of which have utility in the engineering of organisms to target the production of novel fatty acid." How about specifying at least one example?

Is there a prominent place in the paper, perhaps the intro to the discussion, where it says something like residues xx yy zz etc. are primarily responsible for conveying c8-c10 specificity and residues aa bb cc etc. are primarily responsible for C14-C16 specificity? And, changing residues xxx to yyy resulted in a new C4-C6 specificity by shrinking the cavity in conjunction with changing the surface positive charge patch that binds ACP (or whatever).

The authors may be overselling this work when they say: "Bioengineering of the two domains leads to predictable changes." Perhaps they should explain? Being predictable to this reviewer means saying we have done this analysis and are now making the following mutants to test the hypothesis that a predictable outcome will result. Then make the mutants and test the hypothesis. Looking at data and some models may allow you to rationalize function in structural terms, but it doesn't seem to meet the criteria of predictable engineering.

Reviewer #2 (Remarks to the Author):

I believe that the authors have made significant efforts to revise their manuscript. In general, I accept their rebuttal of my review, though I still believe that the manuscript is not particularly easy to follow. But that's more a reflection of the difficulty in describing the complex type of study reported here rather than poor narrative.

Reviewer #3 (Remarks to the Author):

Most of the points raised in my previous review have been addressed by the authors. In addition, please consider the following (minor) points:

Table 1: Please indicate that this table shows free fatty acids in the medium/supernatant, not total fatty acids of the E. coli cells.

Figure 1 c legend: Similar to Table 1, please indicate that the numbers indicate free not total fatty acids. I assume that the blue bars show free fatty acids of the medium, not total E. coli fatty acids

Figure 2 see Figure 1 c: free or total fatty acids?

Suppl. Table 1 legend: domain-swapping not shuffling (two times)

Suppl. Table 2 footnote a: domain-swapping not shuffling

Suppl Table 3: The title is odd:

"Fatty acid titer and fatty acid profiles expressed by site-directed acyl-ACP TE mutants"

Fatty acid titers are not "expressed". and the mutants are not site-directed. The mutations might be site directed. Please indicate that only free fatty acids in the medium are depicted here

The right column shows free not total fatty acids.

Suppl. Fig. 1: Please indicate that only free fatty acids of the medium are shown here.

Suppl. Fig. 2: Please indicate where one can find the explanation for the enzyme abbreviations.

Response to reviewers' comments

Reviewer #1 (Remarks to the Author):

The manuscript is much improved, but I am still frustrated that the abstract and discussion allude to details that have to be dug out of the results section, often via reference to figures or tables. The abstract could be significantly strengthened to include some detail of the domains. Perhaps the domains should be named and those

Response: We've revised the abstract to include more details about the residues that determine the substrate binding cavity and the residues that affect the positively charged surface patches. The detailed residue positions and names are included in the abstract. We thank the reviewer for his patience.

Also authors refer to "Bioengineering of the two domains leads to predictable changes in substrate specificity, many of which have utility in the engineering of organisms to target the production of novel fatty acid." How about specifying at least one example?

Response: In this original sentence, the word "predictable" should be changed to "predicted" to better express what we meant. We did domain swapping, structural modeling, and sequence alignment to generate predictions as to the residues that are responsible for substrate specificity. These were subsequently tested by mutagenesis, which in some cases led to the predicted changes in substrate specificity (i.e., shifting substrate specificity to shorter chain length) and others not. Among all the predicted residues, V194, V217, N223, R226, R227, and I268 of CvFatB2 were proved to directly affect substrate specificity.

Is there a prominent place in the paper, perhaps the intro to the discussion, where it says something like residues xx yy zz etc. are primarily responsible for conveying c8-c10 specificity and residues aa bb cc etc. are primarily responsible for C14-C16 specificity? And, changing residues xxx to yyy resulted in a new C4-C6 specificity by shrinking the cavity in conjunction with changing the surface positive charge patch that binds ACP (or whatever).

Response: In the first paragraph of discussion, we've added text to summarize the residues that are responsible for the C14-16 specificity of CvFatB2 and the residues that are responsible for the C8-10 specificity of CvFatB1.

The authors may be overselling this work when they say: "Bioengineering of the two domains leads to predictable changes." Perhaps they should explain? Being predictable to this reviewer means saying we have done this analysis and are now making the following mutants to test the hypothesis that a predictable outcome will result. Then make the mutants and test the

hypothesis. Looking at data and some models may allow you to rationalize function in structural terms, but it doesn't seem to meet the criteria of predictable engineering.

Response: We understand the reviewer's question is if our model can guide the modification of acyl-ACP TE that will lead to predictable outcomes. To some extent, predictable modification of substrate specificity can be achieved based on the insight we obtained in this study about the structural determinants for the substrate specificity of acyl-ACP TE. For example, residue V217 of CvFatB2 affects the depth of the cavity, and can be changed to a larger hydrophobic residue to shift the substrate specificity to short chain length. We have demonstrated this in our mutagenesis experiments. However, to rationally design a specific chain length selectivity may require additional insights of all the cavity-forming residues and the interaction between TE and the acyl-ACP substrate.

Reviewer #2 (Remarks to the Author):

I believe that the authors have made significant efforts to revise their manuscript. In general, I accept their rebuttal of my review, though I still believe that the manuscript is not particularly easy to follow. But that's more a reflection of the difficulty in describing the complex type of study reported here rather than poor narrative.

Response: We thank the reviewer for his perseverance, and hope that the complexity of the system has been sufficiently explained.

Reviewer #3 (Remarks to the Author):

Most of the points raised in my previous review have been addressed by the authors.

In addition, please consider the following (minor) points:

Table 1: Please indicate that this table shows free fatty acids in the medium/supernatant, not total fatty acids of the E. coli cells.

Figure 1 c legend: Similar to Table 1, please indicate that the numbers indicate free not total fatty acids. I assume that the blue bars show free fatty acids of the medium, not total E. coli fatty acids

Figure 2 see Figure 1 c: free or total fatty acids?

Response: In all cases identified by Reviewer 3#, we've indicated that the data show the free fatty acid titers in the medium.

Suppl. Table 1 legend: domain-swapping not shuffling (two times)

Suppl. Table 2 footnote a: domain-swapping not shuffling

Response: Both omissions have been corrected.

Suppl Table 3: The title is odd:

"Fatty acid titer and fatty acid profiles expressed by site-directed acyl-ACP TE mutants"

Fatty acid titers are not "expressed". and the mutants are not site-directed. The mutations might be site directed. Please indicate that only free fatty acids in the medium are depicted here

The right column shows free not total fatty acids.

Response: The title of Supplementary Table 3 has been changed to "FA composition and total free FA accumulation of *E. coli* strains expressing acyl-ACP TE mutants" We've indicated that the data represent free fatty acids.

Suppl. Fig. 1: Please indicate that only free fatty acids of the medium are shown here.

Response: We've indicated that the data represent free fatty acids in the medium.

Suppl. Fig. 2: Please indicate where one can find the explanation for the enzyme abbreviations.

Response: We've added enzyme abbreviations in the figure legend.